# Influence of Flue Gas Desulfurization Gypsum on Phosphorus Loss in Pine Bark Substrates

Paul C. Bartley [1,*], Landon B. Erbrick [1], Michael J. Knotts [1], Dexter B. Watts [2] and Henry A. Torbert [2]

1  Department of Horticulture, Auburn University, Auburn, AL 36849, USA
2  USDA-ARS National Soil Dynamics Laboratory, Auburn, AL 36832, USA
*  Correspondence: paul.bartley@auburn.edu

**Abstract:** Flue-gas desulfurization (FGD) gypsum, a byproduct of coal-fired electrical utility plants, has been shown to effectively reduce phosphorus (P) leaching in many agricultural systems. However, its applications in horticultural production systems have been insufficiently researched resulting in limited industry adoption. To evaluate the efficacy of FGD gypsum to reduce P leaching in horticultural media, pine bark substrates were amended with FGD gypsum at 2.5, 5, and 10% ($v/v$). In accordance with industry practice, controlled release fertilizer (19N-3P-10K) was amply incorporated into all potting media treatments to support primary nutrient sufficiency of transplanted stock. The greatest P leaching occurred in the control substrates containing only pine bark and fertilizer. The standard pine bark substrate treatment, containing lime and micronutrients, reduced total P leaching by 35% and should be considered a best management practice. The addition of FGD gypsum at 2.5, 5, and 10% ($v/v$) reduced the total P collected in leachate by 47, 59, and 70%, respectively. Gypsum amendments increased potassium leachate concentrations but elevated potassium levels normalized after ~20 days. With little to no effect on substrate physical properties or pH, pine bark substrates can be amended with FGD gypsum to effectively reduce P leaching in short-term crops.

**Keywords:** nursery production; container production; floriculture; growing media; fertilizer; irrigation; amendments

## 1. Introduction

The management practices of horticultural production systems are too often in conflict with the industry's narrative to promote environmental sustainability. One such conflict in the industry concerns eutrophication [1]. Increased nutrient loadings in water systems can disrupt the functions of aquatic ecosystems by reducing water clarity, igniting algal blooms, and creating hypoxic zones [1–4]. Nutrient discharges from agricultural sources have been implicated as the primary cause of diminishing water quality and ecosystem function [5].

To accelerate plant growth, copious quantities of fertilizer are applied and often reapplied to shorten production cycles and drive revenue [6]. Soilless substrates used in container plant production are comprised of materials such as peat and pine bark with low nitrate and phosphate sorption capacities [7,8]. Additionally, these substrate components are engineered or blended to yield physical properties which promote drainage and exacerbate nutrient leaching [9]. Poor irrigation management may lead to nutrient leaching nursery container production [6,10]. A recent survey found that only 20% of container producers in Alabama utilize tools to assist in irrigation management [11]. Compounding the problem further, the concentration of greenhouse and nursery farms is often highest near sensitive wetlands and deltas. Approximately 30% of Alabama's nursery and greenhouse producers are in Mobile and Baldwin counties, home to the second-largest delta in the contiguous United States [12]. These circumstances highlight the need for adaptive management practices that mitigate nutrient loss from horticultural container production systems.

A collective action of research, education, and industry adoption has taken place over the last two decades to promote sustainable practices in container nursery production. Considerable efforts have been made to quantify crop nutrient demands and to promote metered nutrient products such as control-released fertilizers (CRF) [13–17]. Federal support has been appropriated to multi-institutional initiatives such as Clean WateR3, Reduce, Remediate, Recycle (USDA Award # 2014-51181-22372) to assist growers in water management. More recently, substrate additives have been explored to reduce grower susceptibility to excessive nutrient leaching [18,19]. Pine bark is inherently acidic. Therefore, liming is considered a best management practice in nursery production to increase the pH of pine bark within optimal plant nutrient uptake ranges. Shreckhise et al. [18] recorded a 70% cumulative reduction in orthophosphate leaching when pine bark was amended with dolomitic lime and a micronutrient fertilizer. Watts et al. [19] utilized flue gas desulfurization (FGD) gypsum to reduce dissolved reactive phosphorus leaching in a peat:perlite substrate by as much as 75%. These incorporated amendments could provide container plant producers an added layer of protection from environmental conditions such as excessive rainfall.

Despite an increased interest in its agricultural applications in recent decades, FGD gypsum has been investigated only once in horticultural container production applications. Additional research is warranted to determine its effectiveness in abating phosphorus leaching in nursery container production. Therefore, the objective of this study was to investigate P loss from a nursery-grade pine bark substrate amended with FGD gypsum.

## 2. Materials and Methods

### 2.1. Growing Media and Amendments

Nursery-grade pine bark (milled through a 15.9-mm screen) was obtained from Pineywoods Mulch Company (Alex City, AL, USA) on 12 March 2021. Particle size analysis was conducted following Bartley et al. [20]. In summary, three oven-dried 0.5 L samples were passed through a series of 12 sieves (12.5, 9.5, 6.3, 3.35, 2, 1.4, 1, 0.5, 0.3, 0.25, 0.15, and 0.106 mm). The sieves were agitated for 5 min using a Ro-Tap device for agitation. Following agitation, the fractional weight retained on each sieve was recorded. The particle size distribution was expressed as a cumulative distribution. The mean particle size of the pine bark was 2.16 mm with a standard deviation of 0.76 mm. Elemental analysis of the pine bark was determined by Waters Agricultural Laboratories, Inc. (Camilla, GA, USA). Each kg of pine bark substrate was comprised of 5.25 g kg$^{-1}$ N, 1.58 g kg$^{-1}$ P, 15.79 g kg$^{-1}$ K, 8.25 g kg$^{-1}$ Ca, 3.24 g kg$^{-1}$ Mg, 2.47 g kg$^{-1}$ S, 0.09 g kg$^{-1}$ B, 0.05 g kg$^{-1}$ Z, 0.06 g kg$^{-1}$ Mn, 0.32 g kg$^{-1}$ Fe, and 0.01 g kg$^{-1}$ Cu.

FGD gypsum was collected from a local coal-fired electrical utility plant (Alabama Power Gaston Generating Plant, Wilsonville, AL, USA). The material was received as a dry fine powder with a pH of ~7. The elemental composition of the FGD gypsum compared to mined gypsum can be found in Table 1.

**Table 1.** Elemental composition of flue gas desulfurization (FGD) gypsum, a byproduct of coal-fire electrical utility plants, and mined gypsum.

| Element | FGD Gypsum [1] | Mined Gypsum [2] |
|---|---|---|
| Calcium (%) | 21.9 | 24.5 |
| Sulfur (%) | 16.7 | 16.1 |
| Nitrogen (N) (ppm) | Not determined | Not determined |
| Phosphorous (P) (ppm) | 22.3 | 30 |
| Potassium (K) (ppm) | <0.1 | 3600 |
| Magnesium (Mg) (ppm) | 150 | 26,900 |
| Boron (B) (ppm) | 12 | 99 |
| Copper (Cu) (ppm) | 23 | <0.60 |
| Iron (Fe) (ppm) | 327 | 3800 |
| Manganese (Mn) (ppm) | 3 | 225 |
| Nickel (Ni) (ppm) | NA | <0.6 |
| Zinc (Zn) (ppm) | <0.1 | 8.7 |

[1] FGD gypsum collected from Alabama Power Gaston Plant in Wilsonville, AL, USA; [2] Composition of mined gypsum from Dontsova et al. [21].

One day prior to the initiation of the experiment, the pine bark was amended with 4.75 kg m$^{-3}$ of a commercially available 6-month release CRF (Polyon 19-6-12, Harrell's, Lakeland, FL, USA), 0.89 kg m$^{-3}$ granular micronutrient fertilizer (Micromax, Everris, Dublin, OH, USA), and one of five treatment amendments:

1. Neither dolomitic limestone nor FGD gypsum;
2. 4.15 kg m$^{-3}$ dolomitic limestone;
3. 4.15 kg m$^{-3}$ dolomitic limestone and 2.5% (*v/v*) FGD gypsum;
4. 4.15 kg m$^{-3}$ dolomitic limestone and 5% (*v/v*) FGD gypsum;
5. 4.15 kg m$^{-3}$ dolomitic limestone and 10% (*v/v*) FGD gypsum.

From this point forward, each of the five treatments will be referenced by the following (1) Control, (2) Standard, (3) 2.5% FGD Gypsum, (4) 5% FGD Gypsum, and (5) 10% FGD Gypsum. All treatment amendments were incorporated into the substrate by hand-mixing until the samples were adequately homogenized and stored in plastic bags.

### 2.2. Leachate Columns

Twenty-five columns were constructed following Shreckhise et al. [18]. In summary, the columns were constructed of 30 cm sections of PVC pipe (7.8 cm internal diameter). Diffusers and couplers were fixed atop each container to disperse irrigation water evenly. The bottom of each column was fixed with a PVC flat cap with 20 circular holes for drainage. To capture leachate, 0.5 mL beakers were placed under each column (stabilized by a wooden frame).

Five columns were packed with 1.4 L of substrate for each substrate treatment by fixing 30 cm columns to both the top and bottom of each experimental column. The bottom column was capped to retain the substrate. The column stack was loosely filled with substrate and dropped three times on a laboratory bench from a height of ~5 cm to settle the substrate. The top and bottom columns were removed carefully, preserving the substrate in the middle experimental column. The average bulk density for the Control substrate was 0.146 g cm$^{-3}$ and 0.152 g cm$^{-3}$ for the Standard substrate. The 2, 5, and 10% FGD Gypsum treatments had an average bulk density of 0.155 g cm$^{-3}$, 0.166 g cm$^{-3}$, and 0.169 g cm$^{-3}$, respectively. Measured air space, container capacity, and total porosity were 33, 48, and 82%, respectively, according to the NCSU porometer method [22]. The physical properties for all treatments were within 1% of the means reported.

Once packed, the columns were placed in bins and sub-irrigated to saturate the substrate thoroughly. The columns were removed from the bins carefully and allowed to drain for 12 h. The columns were then randomized, vertically mounted, and placed atop a laboratory bench. Each day, for 45 days, the columns were irrigated with 125 mL of tap water [18]. The irrigation water collected periodically over the course of the experiment contained approximately 0.16 ppm P, 2.34 ppm K, 5.99 ppm S, 0.15 ppm Zn, 0.14 ppm Cu, 14.21 ppm Ca, 4.02 ppm Mg, 6.68 ppm Na, and <0.02 ppm B, Mn, Fe, and Al (Waters Agricultural Laboratories, Inc.). On Days 1, 5, 10, 15, 20, 25, 30, 35, 40, and 45, a fraction of leachate was collected and measured for pH and EC (HI9813-6, Hanna Instruments, Woonsocket, RI) and then frozen. Frozen samples were boxed and shipped for analysis (Waters Agricultural Laboratories, Inc.). Total dissolved elemental concentrations were determined with optical emission spectrometry (iCAP 6300 Duo View ICP-OES Spectrometer; Thermo Fisher Scientific, Waltham, MA, USA). The experiment was first initiated on 13 August 2021 and repeated on 2 February 2022. Samples collected on Days 30, 35, and 45 in 2021 were not recovered due to shipping mishandling. As a result, the data presented herein originate primarily from the 2022 run. Data from the 2021 run is presented only where the runs were differentiated.

### 2.3. Statistics

Effects of substrate, time, and the substrate × time interaction on pH, EC, P, K, Ca, S, and Mg were analyzed via two-way repeated measures ANOVA with the PROC Glimmix

procedure of SAS 9.4 (SAS Institute Inc., Cary, NC, USA). An unstructured covariance matrix was utilized. Replications (random) were non-significant ($p > 0.05$) in all ANOVA. Mean leachate concentration of elements significantly influenced by substrate treatment and time interaction ($p < 0.05$) were separated using Tukey's honest significant difference (HSD) at a 5% alpha level by days after initiation. Percent reduction in element concentrations by amended substrates discussed hereafter were calculated based on concentrations in the Control substrate unless otherwise noted.

## 3. Results and Discussion

### 3.1. pH and EC

Leachate pH from amended pine bark substrates was affected by a substrate x time interaction ($p < 0.0001$), which is explained by the precipitous initial increase in pH of substrates containing dolomitic lime compared to the control (Figure 1). The pH of substrates amended with lime increased by ~1 unit over 40 days before a gradual decrease was recorded at Day 45. These ranges are within values reported with comparable lime rates [23], but lower than those reported by Shreckhise et al. [18] in a similar investigation.

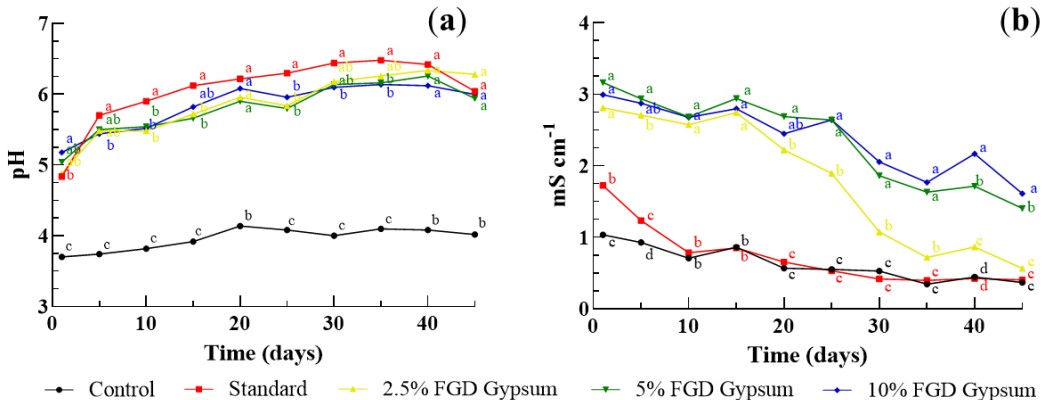

**Figure 1.** The effect of substrate amendments on leachate (**a**) pH and (**b**) electrical conductivity over time in daily-irrigated pine bark columns. Substrate treatments consisted of neither dolomitic limestone nor flue gas desulfurization (FGD) gypsum (Control), 4.15 kg m$^{-3}$ dolomitic limestone (Standard), dolomitic limestone and 2.5% (*v/v*) FGD gypsum, dolomitic limestone and 5% FGD gypsum, or dolomitic limestone and 10% FGD gypsum. All reported values are lsmeans. Treatment effects were analyzed using Tukey's honest significant difference by day. Values containing the same letter were not significantly different ($p > 0.05$).

Coal-fired electrical utility plants can utilize lime slurries to scrub sulfur and other compounds from its flue gas. The resulting chemical interactions precipitate calcium sulfate and gypsum. FGD gypsum often contains <2% residual CaCO$_3$. However, the CaCO$_3$ equivalency of the FGD gypsum utilized in this study was 7.8%. Although this investigation did not observe an increase in pH due to FGD gypsum amendments, the investigators have encountered small increases (~0.5 units) in pH when amended at rates 10% (*v/v*) and higher (data not reported).

Leachate EC was also affected by an interaction between substrate and time ($p < 0.0001$). Substrates containing FGD gypsum recorded the highest EC values on Day 1, 2.81–3.16 mS cm$^{-1}$. These values were significantly higher than the Standard, 1.72 mS cm$^{-1}$, and Control, 1.03 mS cm$^{-1}$, substrates on Day 1. The EC values for FGD gypsum-amended substrates decreased sharply over 45 days. No differences were reported between the Control, Standard, and 2.5% FGD Gypsum treatments on Day 45, suggesting that the lowest amended volume of gypsum had been or was nearly exhausted. Similarly, Watts et al. [19] noted the longevity of similar FGD gypsum-amended volumes over the course of a 9-week assessment. A factor of irrigation volume and leachate fraction, the efficacy of FGD gypsum in long-term container evaluations should be considered in fu-

ture work. The high EC values recorded in FGD gypsum treatments resulted from the dissolution of soluble gypsum. Specifically, Ca, S, and Mg were released in large quantities throughout the study (Table 2). High concentrations of any one element could result in competitive nutrient effects. There has been no literature to date that has investigated this potential interaction following gypsum applications in plant container systems. Before implementing gypsum amendments in a container production system, its effects on floriculture and nursery plant development should be thoroughly studied.

**Table 2.** Effect of substrate amendments on K, Ca, S, Mg over time in leachate of daily irrigated pine bark columns from the 2022 experimental run. Substrate treatments consisted of neither dolomitic limestone nor FGD gypsum (Control), 4.15 kg m$^{-3}$ dolomitic limestone (Standard), dolomitic limestone and 2.5% (*v/v*) FGD gypsum (2.5% FGDG), dolomitic limestone and 5% FGD gypsum (5% FGDG), or dolomitic limestone and 10% FGD gypsum (10% FGDG).

| | Days after Initiation | | | | | | | | | |
|---|---|---|---|---|---|---|---|---|---|---|
| | 1 | 5 | 10 | 15 | 20 | 25 | 30 | 35 | 40 | 45 |
| | Total K mg L$^{-1}$ | | | | | | | | | |
| Control | 84.9a [1] | 78.8c | 59.8b | 64.7ab | 55.3ab | 41.3ns | 46.6ns | 33.9ns | 34.9ns | 35.7ab |
| Standard | 100.2bc | 82.8c | 59.0b | 60.0b | 52.0b | 41.0 | 42.1 | 31.3 | 35.1 | 39.6a |
| 2.5% FGDG | 113.1ab | 103.1b | 87.7a | 84.3a | 64.7ab | 47.8 | 39.1 | 28.6 | 29.9 | 28.6b |
| 5% FGDG | 135.2a | 120.3a | 88.1a | 87.1a | 74.0a | 52.9 | 41.7 | 33.3 | 29.3 | 31.8ab |
| 10% FGDG | 122.4ab | 112.5ab | 88.4a | 81.4ab | 62.5ab | 51.7 | 47.9 | 34.6 | 33.0 | 32.7ab |
| *p* value | 0.0004 | <0.0001 | 0.0070 | 0.0146 | 0.0496 | 0.0807 | 0.1724 | 0.4706 | 0.4592 | 0.0492 |
| | Total Ca mg L$^{-1}$ | | | | | | | | | |
| Control | 28.9d | 24.5d | 14.5c | 16.1b | 13.2c | 6.9c | 8.8c | 5.4c | 5.3d | 6.1c |
| Standard | 126.8c | 77.1c | 40.3c | 37.2b | 28.7c | 17.9c | 19.2c | 12.6c | 15.3d | 24.3c |
| 2.5% FGDG | 468.8b | 439.8b | 444.2b | 442.2a | 380.1b | 301.5b | 188.2b | 119.9b | 117.3c | 92.6b |
| 5% FGDG | 500.3a | 480.8a | 483.1a | 484.2a | 487.3a | 491.3a | 449.1a | 425.3a | 356.3b | 396.8a |
| 10% FGDG | 504.7a | 493.2a | 484.5a | 482.6a | 483.5a | 477.2a | 488.7a | 466.6a | 476.5a | 457.6a |
| *p* value | <0.0001 | <0.0001 | <0.0001 | <0.0001 | <0.0001 | <0.0001 | <0.0001 | <0.0001 | <0.0001 | <0.0001 |
| | Total S mg L$^{-1}$ | | | | | | | | | |
| Control | 44.3c | 38.9d | 26.8b | 28.6b | 25.5c | 17.7c | 20.9c | 15.5c | 15.8d | 17.2c |
| Standard | 203.7b | 118.8c | 54.0b | 42.7b | 27.4c | 18.6c | 18.9c | 13.6c | 15.7d | 17.7c |
| 2.5% FGDG | 606.7a | 543.9b | 524.8a | 500.8a | 401.2b | 301.5b | 177.3b | 112.8b | 106.7c | 68.7c |
| 5% FGDG | 640.8a | 603.3a | 565.3a | 546.1a | 518.3a | 499.2a | 425.3a | 394.2a | 318.2b | 315.3b |
| 10% FGDG | 641.0 | 605.0 | 563.1a | 533.3a | 497.2a | 478.4a | 474.6a | 433.7a | 433.0a | 380.5a |
| *p* value | <0.0001 | <0.0001 | <0.0001 | <0.0001 | <0.0001 | <0.0001 | <0.0001 | <0.0001 | <0.0001 | <0.0001 |
| | Total Mg mg L$^{-1}$ | | | | | | | | | |
| Control | 21.9c | 18.6d | 10.8b | 11.6b | 9.1c | 4.8c | 5.9c | 3.5c | 3.5d | 4.1d |
| Standard | 83.1b | 47.9c | 24.0b | 22.3b | 17.2c | 10.8c | 11.5c | 7.5c | 9.2d | 14.8cd |
| 2.5% FGDG | 166.2a | 134.7b | 110.3a | 101.1a | 68.1b | 43.2b | 25.5b | 16.2b | 15.9c | 12.0bc |
| 5% FGDG | 176.6a | 152.9a | 116.6a | 108.5a | 85.4a | 62.2a | 42.1a | 32.6a | 23.1b | 21.9ab |
| 10% FGDG | 174.3a | 146.3ab | 112.6a | 98.4a | 72.7ab | 61.5a | 51.6a | 34.1a | 29.4a | 26.6a |
| *p* value | <0.0001 | <0.0001 | <0.0001 | <0.0001 | <0.0001 | <0.0001 | <0.0001 | <0.0001 | <0.0001 | <0.0001 |

[1] Treatment effects were analyzed using Tukey's honest significant difference by day (column). Values containing the same letter were not significantly different (*p* > 0.05).

### 3.2. Phosphorus

Total dissolved P concentrations were also affected by an interaction between substrate and time (*p* < 0.0001). The highest P concentrations were recorded in the Control substrate (Figure 2a). Standard and gypsum- amended substrates significantly reduced the P leached from pine bark columns through Day 40. At Day 45, no significant differences were recorded across treatments. In the 2022 run of the study, FGD gypsum did not significantly improve P retention greater than the Standard (lime-amended substrate).

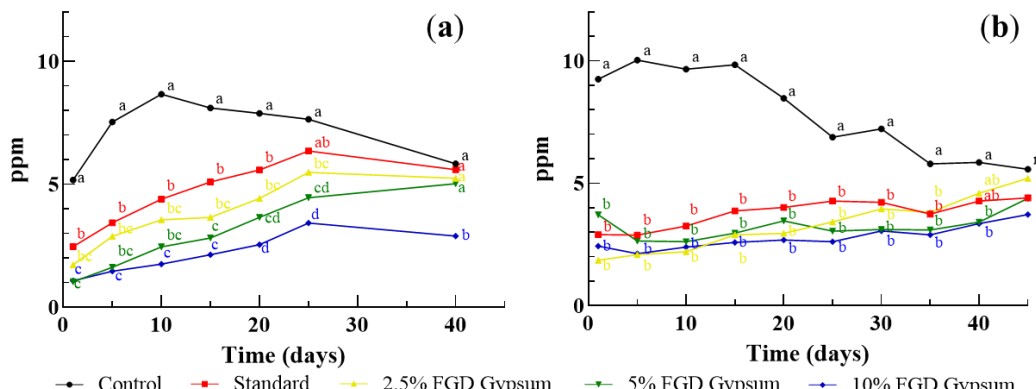

**Figure 2.** The effect of substrate amendments on total P leaching over time in daily irrigated pine bark columns container conducted in (**a**) 2021 and (**b**) 2022. Substrate treatments consisted of neither dolomitic limestone nor flue gas desulfurization (FGD) gypsum (Control), 4.15 kg m$^{-3}$ dolomitic limestone (Standard), dolomitic limestone and 2.5% (*v/v*) FGD gypsum, dolomitic limestone and 5% FGD gypsum, or dolomitic limestone and 10% FGD gypsum. Due to mishandling of samples in 2021, data for Day 30, 35, and 45 were unrecovered. All reported values are lsmeans. Treatment effects were analyzed using Tukey's honest significant difference by day. Values containing the same letter were not significantly different ($p > 0.05$).

These results contrast with the data recovered in 2021 (Figure 2b). In the first experimental run, P retention was incrementally improved by increasing FGD gypsum amendments. The 10% FGD Gypsum substrate lost an average of 0.27 mg of P per day, 70% less than the Control substrate and 54% less than the Standard substrate through 40 days. A diminishing efficacy in P retention was observed in both experimental runs. These results contrast with Watts et al. [19], where a significant improvement of P retention in gypsum amended, peat-based substrates was reported through 60 days. However, in that study, the P concentrations were an order of magnitude greater, potentially due to the initial P charge, the type of CRF utilized, and the irrigation applied. The diminishing efficacy observed in this study could be the result of a combination of factors, such as a reduced P availability, observed by the decline in P concentration in the Control substrate, or the threshold of a Ca–P dissociation equilibrium, evidenced by the lack of change in P leachate concentration in amended substrates in 2022. While it is unclear why all trends were not replicated in both experimental runs, these results highlight the potential for FGD gypsum to remediate P leaching concerns in nursery container production.

These results also support the use of dolomitic lime as a best management practice to mitigate P leaching in container production [13,18]. Evidence of P precipitation or adsorption of P ions by liming agents has also been observed in previous studies [24,25]. Argo and Biernbaum [24] reported that the amount of soluble P was inversely proportional to the amount of liming agent amended to peat-based substrate. Shreckhise et al. [18] observed similar results in a pine bark substrate. These reductions were speculated to be the result of $CaHPO_4$ or $CaH_5O_6P$ precipitates. Reductions in P leaching as a result of FGD gypsum amendments could be the result of similar Ca–P complexes.

The growth, vigor, and maturation rate of horticulture plants are of upmost importance to the grower. Additionally, soilless substrates used in the floriculture and nursery industry have limited P holding capacity [26–28]. These facts may lend justification for augmented P fertilizer applications. Since only orthophosphates are available for plant uptake, P bound in Ca–P complexes may be unavailable during critical stages of plant development and, thus, detrimental to the grower. Conversely, retaining P in Ca–P complexes within the substrate could extend its availability, buffering P deficiencies. As noted by Henry et al. [29], P deficiency may occur in as little as 3 weeks. The lability of P during crop development with respect to FGD gypsum applications should be further investigated with a wide range of horticultural plants.

### 3.3. Potassium, Calcium, Sulfur, Magnesium, and Aluminum

Potassium leachate concentrations were influenced by a substrate and time interactions ($p < 0.0001$). The potassium concentrations were initially highest in FGD gypsum-amended substrates (Table 2). At Day 10, K leachate concentrations were ~30× higher in the 10% FGD Gypsum substrate relative to the Control substrate. However, from Day 25 to 45, few differences were recorded across all treatments. FGD gypsum, unlike mined gypsum, contains modest amounts of potassium (Table 1). The additional charge of K in gypsum amended substrates could have resulted in the initial spike recovered in the leachate. In field assessments, gypsum has been shown to reduce exchangeable K, indicating that soil-bound K is more susceptible to leaching following amelioration [30]. Additional studies have recorded similar interactions with Mg [31] and Fe [32], emphasizing the need to investigate gypsum-induced nutrient leaching effects. These results may warrant investigations into more metered applications of K in nursery production.

Calcium concentrations were affected by a substrate and time interaction ($p < 0.0001$). Solution Ca concentrations were significantly higher in FGD Gypsum treatments compared to the Standard and Control treatments (Table 2). For example, the 5% FGD Gypsum leachate Ca concentrations were 29 times higher than that of the Standard substrate on Day 25. While Ca concentrations across FGD Gypsum treatments were similar on Day 15, the 2.5% FGD Gypsum leachate contained ~80% less Ca than the other FGD Gypsum treatments at Day 45. The EC data (Figure 1) parallels the concentration of Ca in the 2.5% FGD Gypsum treatment and provides additional support that the gypsum was at or near exhaustion by Day 45. Concentrations of Ca in the 5% FGD Gypsum and the 10% FGD Gypsum treatments were not significantly different at any point during the study. The 5% FGD Gypsum and 10% FGD Gypsum leachate Ca concentrations declined by 21% and 9%, respectively, from Day 1 to Day 45. The longevity of gypsum efficacy may be of particular interest in nursery container production where plants may experience multiple seasons and nutrient applications within the same container. While higher concentrations of gypsum may not be advisable, future work could focus on quantifying the longevity of gypsum efficacy beyond the range evaluated here or alternative methods of application to replenish dissolved gypsum, such as top-dressing container substrates.

Magnesium concentrations were also affected by a substrate and time interaction ($p < 0.0001$). Leachate concentrations of Mg in FGD Gypsum treatments were 8× and 2× higher than the Control and Standard treatments, respectively, at Day 1. Magnesium concentrations declined rapidly by the conclusion of the study. Magnesium concentrations in the 10% FGD Gypsum substrate leachate declined to a minimum concentration of 27 ppm, an 85% reduction in 45 days. Magnesium concentrations have been shown to have strong correlation with Ca concentrations [18]. Similarly, the sorption of P in the form of Mg surface complexes has been speculated [33].

Of the elemental concentrations recorded in this study, S concentrations were greatest. Concentrations exceeding 600 ppm S were observed in all FGD Gypsum treatments at Day 1. These were significantly higher than Standard and Control substrates which contained 204 ppm and 44 ppm S, respectively. While the concentration of S decreased in the leachate over 45 days, 5% FGD Gypsum and 10% FGD Gypsum treatments sustained considerably high concentrations of S (>300 ppm). Sulfates, often applied as ammonium sulfate or iron sulfate, have been shown to decrease and suppress soilless container substrate pH long-term [34,35]. However, during the span of this relatively short-term study, pH was unaffected by considerable sulfur charges. It is possible that the relatively high calcium carbonate levels in the tested FGD gypsum adequately suppressed the consequences of S oxidation in this study.

Although gypsum may be applied to reduce the incidence of Al toxicity in some soils [36,37], such challenges with Al and other metal ions are rarely of concern in soilless container substrates. Control treatments had <0.01 ppm Al in the leachate. Standard treatments were marginally higher at 0.2 ppm Al at Day 1 before quickly declining to undetectable levels by Day 10. Treatments containing FGD Gypsum averaged 2.4 ppm

Al in the leachate from Day 1 to Day 25 before declining to undetectable levels. Aluminum sulfate applications are critical to produce blue inflorescences in container-grown hydrangea [38,39]. Pore-water concentrations at ~0.3 ppm Al were shown to be effective at altering sepal color [40]. The Al levels detected in this study in FGD Gypsum treatments were also lower than the upper limits of the recently revised EPA Al freshwater toxicity level, 4.8 ppm. These results suggest that FGD gypsum could be applied to supplement Al in the production of blue hydrangea inflorescence. However, more research is needed to investigate the timing and efficacy of FGD gypsum use in this manner.

## 4. Conclusions

Though familiar in many aspects of agriculture, applications of gypsum have received little support for adoption in container plant production. This research highlights the potential for FGD gypsum, a byproduct of the utility industry, to potentially remediate P leaching without changing the physical characteristics of the substrate. When compared to non-amended substrates, pine bark amended with FGD gypsum reduced P leaching by 59–70%. The effects of FGD gypsum were diminished by conclusion of the study (45 Days after initiation). The longevity, extent of P-retention, and alternative methods of application need to be further investigated. This work also brings to light knowledge gaps regarding its potential utilization as a nutrient supplement or antagonist in container plant production systems. Additional work is required to further understand the impact of FGD gypsum across the expansive variety of horticultural plants produced in the floriculture and nursery industry. If gypsum additions to pine bark and alternative soilless substrates can be accomplished without phytotoxic effects, FGD gypsum may be considered an adaptive management practice to reduce P leaching in container plant production.

**Author Contributions:** Conceptualization, D.B.W. and P.C.B.; methodology, D.B.W. and P.C.B.; software, P.C.B.; validation, P.C.B. and L.B.E.; formal analysis, P.C.B.; investigation, L.B.E.; resources, H.A.T.; data curation, L.B.E. and M.J.K.; writing—original draft preparation, P.C.B.; writing—review and editing, P.C.B.; visualization, P.C.B., L.B.E. and M.J.K.; supervision, P.C.B.; project administration, H.A.T. All authors have read and agreed to the published version of the manuscript.

**Funding:** This research received no external funding.

**Institutional Review Board Statement:** Not applicable.

**Data Availability Statement:** The data presented in this study are available on request from the corresponding author.

**Acknowledgments:** The authors would like to thank USDA-NIFA for their generous support of our continued research and Southern Company for their assistance in acquiring research materials.

**Conflicts of Interest:** The authors declare no conflict of interest.

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
