# Peer review of "Influence of Flue Gas Desulfurization Gypsum on Phosphorus Loss in Pine Bark Substrates"

_agriculture, doi:10.3390/agriculture13020283_

Round 1
Reviewer 1 Report
The manuscript mainly proves that.“With little to no effect on substrate physical properties or ph, Fgd gypsum can beamended with pine bark substrates to effectively reduce P leaching in short-term crops”. In order to better display the research content and improve the quality of the manuscript, it is recommended that the author check the manuscript more thoroughly. In order to improve the manuscript, several problems need to be further solved.
1. It is suggested to classify the materials and methods after integration, and add subtitles to make the logic clear and organized.
2. What test is used for data calculation in Figure 1? What are data statistics software and analysis software?
3. Do all experiments involved in the whole experiment have biological or technical repetitions set up? How many repetitions are there? What is the reliability of the data obtained? All data problems are the same as above.
4. This paper draws such a conclusion that the specific meaning can be described in detail, and the results can be extended to a certain extent to make the content of the article more reasonable, plump and more extensible.
5.Whether the format of the secondary heading in line 141 is correct, please check and revise the whole text.
6.The content of the discussion section is too little, and it is suggested that the relationship between Flue-gas desulfurization (FGD) gypsum and metal ion action should be added appropriately.
7.The content of the conclusion part is not enough to highlight the research significance of the manuscript.
8.The proportion of references within 3-5 years is too little less than 30%, and many references 20 years ago, which makes people doubt whether the applicability of the study content is too low.
Author Response
Thank you for your time and suggestions to improve our article.

Reviewer 2 Report
1. In the line no. 69 of the manuscript, authors mentioned "In summary, three oven-dried samples, 0.5 L". What authors mean by 0.5 L? In which form the authors have used FGD gypsum in the present study?
2. In line no. 189-191 the authors mentioned that "Standard and gypsum amended substrates significantly reduced the 189 P leached from pine bark columns through Day 40. At Day 45, no significant differences 190 were recorded across treatments."
It would be interesting if the authors could mention the plausible reason for such trend.
Author Response

(The authors gave the same response as above.)

Reviewer 3 Report
Review of: Influence of FGD gypsum on phosphorus loss in pine bark substrates
The manuscript describes a repeated experiment that quantified leachate quality from simulated nursery containers prepared by fertilizer-amended pine park substrate with or without dolomitic-lime supplementation. Dolomitic-lime supplemented substrate was further amended with 2.5 to 10% FGD gypsum by volume. Leachate quality was collected and analyzed 5- to 45-days after initiation.
The work is novel and practical and features a straightforward writing style. I think the amount of experimental data presented just meets the minimum threshold for a full-length research article, but it may make a better research brief, or short communication, or whatever MDPI refers to these things as. It should be revised for clarity; specifically in regard to ‘substrate’ usage. I’m not insisting, but I do encourage the authors to replace ‘substrate’ with ‘potting media’ or ‘growth media’ throughout the manuscript. Some parts of the manuscript should be revised for accuracy, and the results perhaps synthesized further.
Specific comments:
L.8, consider replacing ‘waste product’ with ‘byproduct’?
L.13, consider revising to ‘In accordance with industry practice, controlled release fertilizer (19N-3P-10K) was amply incorporated into all potting media treatments to support primary nutrient sufficiency of transplanted stock.’ This revised description of the fertilizer content is not a type-o. Per fertilizer grade reporting convention, the 19-6-12 Polyon contains 6% P2O5 and 12% K2O, equating to 2.6% elemental P and 10% elemental K, respectively. Please revise to elemental mass content throughout the manuscript....but the first mention of the actual commercial 19-6-12 fertilizer product (L.87-88) should remain as is.
L.19-20, interchange ‘FGD gypsum’ and ‘pine park substrates’ in this last sentence.
L.31-32, I would omit this sentence.
L.33-34, I think ‘crops’ is a misnomer here and most readers recognize container production as a business. Consider rewording this sentence.
L.50, this is a good spot to introduce the CRF acronym.
L.60, ‘a byproduct of coal burning power plants’ can be safely omitted.
L.74, do you mean supplement 1? I think Figure 1 is something different.
L.76-78, consider revising to ‘Each kg of pine bark substrate was comprised of 5.25 g N, 1.58 g P, etc.’
L.84+, if this mined gypsum elemental composition comes from a reference, please cite it.
L.105, consider 1.4 L instead of 1,400 cm3
L.108, 2 in should be converted to SI units
L.134 ‘to determine treatment effects’ can be safely omitted.
L.135-137, I don’t like when reviewers re-write my manuscripts, but consider revising this sentence as ‘Mean leachate concentration of elements significantly influenced by substrate treatment and time interaction (p<0.05) were separated using Tukey’s Honest Significant Difference (HSD) at a 5% alpha level by days after initiation.’ This is merely a suggestion. Likewise, reporting of least squares means is standard, thus this sentence can be safely omitted.
L.151-154, this is meaningful information that should be moved to the Introduction.
L.154-156, perhaps this can be revised to say: ‘While not specifically determined here, FGD gypsum often contains <2% residual CaCO3. However, the combination of FGD gypsum and dolomitic limestone within the described experimental treatments precludes the likelihood of residual influence on substrate leachate chemistry.’ Or something contextually similar.
L.157-162, the exact rates of complete and micronutrient fertilizer can be safely omitted from the Figure 1 caption. It is imperative that the letters superimposed on the figure be identified; i.e., do they represent the HSD statistical groupings at a 5% alpha within each DAI? Likewise, a sentence describing what the error bars represent must be added, or the error bars should be omitted. If the error bars represent the SE of each treatment level mean at each DAI, then please omit them. Standard errors of the mean afford zero statistical inference.
L.180-184, please indicate whether these are means pooled over both experimental runs, or something different. Given the P content of leachate is depicted in Figure 2, the P data should be omitted from this table. The exact rates of complete and micronutrient fertilizers aren’t needed in this Table caption. The exact rates of complete and micronutrient fertilizer can also be safely omitted from the Figure 1 and 2 captions.
L.185+, I suggest rounding up these leachate concentrations to show only one significant digit right of the decimal point.
L.201-207, the exact rates of complete and micronutrient fertilizer can be safely omitted from the Figure 2 caption. It is imperative that the letters superimposed on the figure be identified; i.e., do they represent the HSD statistical groupings at a 5% alpha within each DAI? Likewise, a sentence describing what the error bars represent must be added, or the error bars should be omitted. If the error bars represent the SE of each treatment level mean at each DAI, then please omit them.
L.221, consider revising to ‘Though familiar in many....’
L.226-228, please revise for clarity.
Either in the 3.2 Phosphorus subsection of Results or in the Conclusions section, the authors should describe the mechanism behind FGD-induced reduction of P solubility. This description will likely rely heavily on the agricultural literature. Given vigor and rapid maturation of the horticultural specimen transplanted into the container is the grower’s primary interest, and justifies copious fertilizer incorporation into container media entirely, the absence of plants form the described experiment is notably conspicuous. I strongly encourage the authors to acknowledge this missing half with concern for this FGD-induced reduction of P solubility side-effect, and to identify future research directions that may also ensure critical plant-availability of phosphorus during early-stage propagation.
Author Response

(The authors gave the same response as above.)
